# Epidemiology of Cognitive Impairments: Demographic and Clinical Predictors of Memory and Attention Challenges—Findings from Twelve National Disability Indicators

**DOI:** 10.3390/jcm14186390

**Published:** 2025-09-10

**Authors:** Ahmed Alduais, Hind Alfadda, Hessah Saad Alarifi

**Affiliations:** 1Department of Psychology, Norwegian University of Science and Technology, NO-7491 Trondheim, Norway; 2Department of Curriculum and Instruction, College of Education, King Saud University, Riyadh 11362, Saudi Arabia; 3Department of Educational Administration, College of Education, King Saud University, Riyadh 11362, Saudi Arabia; arifi-hs@ksu.edu.sa

**Keywords:** cognitive difficulties, memory impairment, concentration, gender disparities, regional inequities, national disability survey, Saudi Arabia

## Abstract

**Background**: Cognitive difficulties involving memory and concentration significantly affect individuals’ daily functioning and quality of life, influenced by demographic, clinical, and socio-environmental factors. This study aimed to examine the national prevalence and distribution of cognitive difficulties in Saudi Arabia, explore regional and gender disparities, and identify demographic and clinical predictors. **Methods**: Data were obtained from the 2017 Saudi National Disability Survey, a population-based, cross-sectional study involving 20.4 million Saudi citizens. Twelve indicators related to cognitive difficulty—covering severity, educational and marital status, consanguinity, duration, causes, and regional distribution—were analyzed, integrating baseline population data from three national surveys. **Results**: Results indicated that 1.1% (224,408 individuals) reported cognitive difficulties, predominantly alongside other disabilities (1.0%), while only 0.2% reported it exclusively. Cognitive difficulties were significantly higher in Al-Riyadh and Makkah Al-Mokarramah, with residents of Aseer and Hail facing doubled odds compared to Najran. Gender disparities were evident; males predominated in extreme severity and congenital or accident-related cases, whereas females showed higher proportions of disease-related causes, mild severity, and prolonged disability (≥25 years). Independent predictors included severe and extreme severity, disease-related causes, consanguinity, and long duration. **Conclusions**: These findings highlight critical regional and gender-based inequities and underscore the need for targeted policies emphasizing early detection, gender-sensitive interventions, and region-specific resource allocation to meet Saudi Arabia’s Vision 2030 objectives for inclusive health and social services.

## 1. Introduction

Cognitive difficulties involving memory and concentration manifest across a spectrum of intensity, from mild lapses in daily functioning to severe impairments that significantly disrupt an individual’s ability to engage in routine activities [1,2]. These difficulties are influenced by a range of medical, psychological, and environmental factors, and their severity often correlates with underlying health conditions and cognitive reserves [3,4,5,6]. This paper provides a detailed analysis of the prevalence and epidemiology of cognitive difficulty in Saudi Arabia, utilizing data from the national disability survey. To establish a comprehensive context for this analysis, the following sections will review the nature of cognitive difficulties, their underlying causes, global and local prevalence, and the existing support systems within the Kingdom. Cognitive impairment is commonly defined as a reduction in memory, concentration, or both, that interferes with everyday activities and exceeds normal age-related changes. In population surveys, it is measured using the Washington Group Extended Set, which asks, “Do you have difficulty remembering or concentrating?” with graded response categories from mild to extreme difficulty [7,8].

## 2. The Nature and Manifestation of Cognitive Difficulties

Cognitive difficulties most commonly affect memory and attention, which are critical for daily functioning and higher-order cognition. Deficits in these domains manifest across diverse populations. For instance, individuals with acquired brain injury frequently experience memory problems due to disruptions in attention, memory, and executive functions, impairing their ability to recall personal experiences or perform daily tasks [9]. Similarly, in mild cognitive impairment (MCI), subjective memory complaints are common and often correspond to objective memory deficits and structural brain changes, such as reduced hippocampal thickness, with depressed mood emerging as a strong predictor of perceived decline [10,11]. Postmenopausal women also report subjective memory impairments that are significantly associated with poorer performance on standardized assessments [12]. Attention problems further compound these challenges; individuals with insomnia show impairments in working and episodic memory [13], while children with chronic fatigue syndrome struggle to sustain or switch attention, negatively impacting academic performance [14]. Likewise, children and adolescents with dissociative disorders display poor cognitive functioning, particularly in verbal working memory and sustained attention [15].

At the neurological level, working memory capacity plays a pivotal role in fluid intelligence, reasoning, and academic achievement, with variations linked to differences in attention control and stress resilience [16,17]. Impaired attention and cognitive control can precede errors in working memory tasks [18], while sustained attention is closely tied to cognitive energy regulation [19]. Psychological factors such as anxiety and depression reduce memory efficiency by impairing concentration [9,20], and environmental stressors such as noise further tax cognitive resources [21,22]. Medical and lifestyle influences also contribute: sleep disruption, chronic illness, and inactivity worsen memory and attention, whereas interventions like exergaming show potential to improve working memory and attention performance [23]. Collectively, these findings demonstrate that memory and concentration difficulties are shaped by an interplay of neurological, psychological, medical, and lifestyle-related factors.

### 2.1. Etiology and Contributing Factors

Underpinning these varied manifestations, memory and concentration difficulties arise from a range of neurological, psychological, medical, and lifestyle-related factors. Neurological conditions such as acquired brain injury, epilepsy, and neurodegenerative diseases like dementia are primary contributors [9,24,25]. Psychological factors, including anxiety, depression, and stress, also significantly impair attention and reduce memory efficiency [9,26]. Furthermore, medical factors such as chronic pain, long-term opioid use, and chemotherapy are associated with cognitive deficits [27,28], while aging naturally affects cognitive resources [29].

Among these causes, lifestyle factors play a pivotal role in both exacerbating and mitigating cognitive challenges. Chronic stress, poor sleep quality, and sedentary behavior are strongly linked to impaired attention and memory [30,31]. Conversely, regular physical activity, mentally stimulating activities, and strong social engagement are associated with better cognitive outcomes and resilience [32,33]. These modifiable factors highlight the importance of holistic, preventive strategies in maintaining cognitive health across the lifespan.

Pathophysiological mechanisms underlying cognitive impairment include neurodegeneration, vascular compromise, endocrine and metabolic dysfunction, and systemic inflammation. For example, hippocampal atrophy and amyloid deposition contribute to memory loss, while cerebrovascular disease and chronic hypoxia reduce attentional capacity [27,34]. These pathways interact with psychological stress and environmental exposures, compounding risk. Recent interventional studies further suggest that targeted lifestyle strategies may mitigate decline. For example, Baduanjin exercise was shown to improve lipid profile, blood pressure, and thyroid function in women with subclinical hypothyroidism and MCI, highlighting the role of modifiable factors [35].

### 2.2. Prevalence of Cognitive Impairment: Global and Local Perspectives

The widespread influence of these risk factors contributes to a significant global burden of cognitive difficulties, particularly among middle-aged and older adults. Harmonized data from 11 cohorts place MCI prevalence between 3% and 11% [36], while a global review reported a median prevalence of 19% in community studies of adults aged ≥ 50 years [4]. In Europe, rates of 15.5% in Portugal and 18.5% in Spain have been recorded in older adults, with higher rates in women and the oldest age groups [3,37]. Similarly, data from Mexico City shows impairment climbing from 10% at ages 50–59 to 55% at 80–89 [38]. Psychiatric comorbidity is also a key factor, with major depression yielding the greatest population burden [34], underscoring the need for systematic screening. It should be noted that these prevalence estimates are not restricted to the MCI alone; cognitive difficulties may also arise from conditions such as depression, schizophrenia, multiple sclerosis, and Parkinson’s disease, all of which carry higher risk of cognitive deficits [34].

Globally, the prevalence of MCI varies between 3% and 11% in harmonized cohort data [36], with systematic reviews reporting a median prevalence of 19% among adults aged 50 years and older [4]. In Europe, prevalence estimates range from 15.5% in Portugal to 18.5% in Spain, with higher rates among women and the oldest groups [3,37]. In Mexico City, prevalence increased from 10% among adults aged 50–59 to 55% among those aged 80–89 [38]. In Saudi Arabia, community screening in Riyadh found cognitive impairment in 45% of participants [39], while a primary care survey reported prevalence of 21% [40].

Reflecting these global trends, cognitive impairment is also widespread among older Saudis. Community screening in Riyadh detected deficits in 45% of participants [39], and primary care surveys found impairment in 21% of clinic attendees [40]. Key risk factors in this population include age, low education, cardiovascular disease, polypharmacy, depressive symptoms, and poor physical performance [2]. Furthermore, 6.1% of community-dwelling elders meet the criteria for cognitive frailty [41]. Despite this high prevalence and the significant burden of comorbid mental disorders [42,43], clinical recognition remains weak, highlighting the urgent need for early screening protocols [1,44].

### 2.3. The Landscape of Cognitive Difficulty in Saudi Arabia

The high prevalence of cognitive difficulties among Saudis arises from a complex interplay of chronic disease, genetics, mental health, lifestyle, and socioeconomic status. Hypoxia-linked chronic obstructive pulmonary disease commonly coexists with cognitive deficits, yet screening is rare [44]. Conditions like multiple sclerosis, hypertension, and cardiovascular disease are also major contributors to cognitive impairment [39,45]. Genetic studies have identified both Alzheimer-related loci and rare variants causing intellectual disability [46,47]. These biological predispositions are magnified by lifestyle factors like heavy amphetamine use and excessive smartphone time [48,49], as well as socioeconomic factors such as low education and income [1,40].

In response to this growing public health challenge, Saudi Arabia offers various supports for people with cognitive difficulties, yet fragmentation and training gaps persist. While education programs mandate accommodations, implementation is hampered by limited teacher preparation [50,51]. Vocational rehabilitation services are restricted by bureaucracy and stigma [52], and within healthcare, routine screening for cognitive impairment is uncommon due to a lack of training and protocols [1,44]. Although some progress has been made with assistive technology and family-centered support [41,53], significant gaps remain in creating a coordinated, evidence-based system of care.

Well-documented risk factors for cognitive impairment include advanced age, lower educational attainment, cardiovascular disease, polypharmacy, depressive symptoms, and poor physical performance [1,2]. In Saudi Arabia, consanguinity is also an important determinant, particularly first-degree parental relationships, which increase genetic vulnerability [47].

### 2.4. The Present Study

While previous studies have provided valuable insights into cognitive impairment in specific Saudi contexts, such as community screenings in Riyadh [39] and primary care settings [40], a comprehensive national overview has been lacking. Much of the existing research is localized or focused on specific clinical populations, leaving a critical gap in understanding the full epidemiological landscape across all 13 administrative regions of the Kingdom. This study directly addresses this gap by analyzing the 2017 national disability survey, a large-scale, population-based dataset. The rationale for this approach is to move beyond localized findings to construct the first detailed, nationally representative map of cognitive difficulty in Saudi Arabia. This analysis is essential for identifying high-risk regions and demographic groups, thereby providing policymakers and healthcare planners with the robust evidence needed to develop targeted interventions and allocate resources effectively.

The aims of this study are threefold: (1) to describe the prevalence and distribution of cognitive difficulty across 13 administrative regions and demographic groups; (2) to examine sex differences in the experience and reporting of cognitive difficulty using chi-square tests across all 12 indicators; and (3) to identify independent predictors of cognitive difficulty through multivariable logistic regression. These aims are supported by a series of six analytical tables that present descriptive, bivariate, and multivariable findings, offering a robust evidence base for public health planning, early intervention strategies, and future research on cognitive health in the Kingdom.

## 3. Methods

### 3.1. Design

This study employed a cross-sectional, population-based design using secondary data from the 2017 Disability Survey conducted by the General Authority of Statistics (GAStat) in Saudi Arabia [54]. The survey was designed to provide nationally representative estimates of disability prevalence and associated characteristics across all 13 administrative regions. The cross-sectional nature of the survey allowed for the assessment of the prevalence and distribution of cognitive difficulty at a single point in time, making it suitable for descriptive and inferential epidemiological analysis.

The use of secondary data from a nationally representative survey offers several methodological advantages. First, it ensures high external validity due to the large sample size (N = 20.4 million Saudis) and rigorous two-stage stratified cluster sampling design, minimizing selection bias and supporting generalizability [55]. Second, secondary data analysis promotes research efficiency and transparency by reusing high-quality, publicly available data, reducing duplication of effort and costs [56]. Third, national disability surveys like this one are essential for monitoring progress toward the Sustainable Development Goals (SDGs), particularly SDG 3 (Good Health and Well-being) and SDG 10 (Reduced Inequalities), by providing disaggregated data on vulnerable populations [57].

Furthermore, the two-stage stratified cluster sampling design, based on the 2010 Population and Housing Census, ensured proportional representation across regions, urban-rural strata, and demographic groups [54]. This design is widely recommended for large-scale household surveys because it enhances precision and supports valid statistical inference [55].

### 3.2. Sample

The study utilized data from the 2017 Disability Survey, conducted by the GAStat in Saudi Arabia [54]. The survey employed a two-stage stratified cluster sampling design to ensure national representativeness across all 13 administrative regions. In the first stage, Primary Sampling Units (PSUs) were selected from the updated framework of the 2010 Population and Housing Census [54]. A total of 1300 PSUs were randomly drawn out of 3600 available PSUs, distributed proportionally across regions. In the second stage, up to 25 households were systematically sampled from each PSU, resulting in a final sample size of 33,575 households nationwide.

For this study, 12 indicators related to cognitive (memory and concentration) difficulty were extracted from the 2017 Disability Survey. These indicators included severity levels (mild, severe, extreme), gender distribution, educational status, marital status, consanguinity, cause of disability, duration of disability, and use of sign language. Additionally, three baseline datasets from the 2017 Population Characteristics Survey were incorporated to calculate regional prevalence rates and ensure national representativeness [54].

### 3.3. Measures

This study utilized 12 key indicators related to cognitive (memory and concentration) difficulty, derived from the 2017 Disability Survey conducted by GAStat [54]. This national survey adopted the Washington Group Extended Set on Functioning (WG-ES), an internationally validated instrument aligned with the World Health Organization’s International Classification of Functioning, Disability, and Health (ICF) [7]. The WG-ES includes standardized questions on cognitive functioning, specifically assessing attention and memory using items such as “Do you have difficulty remembering or concentrating?” with response categories of no difficulty, some difficulty, a lot of difficulty, or cannot do at all. The survey elements are also consistent with the WHO Disability Assessment Schedule 2.0 (WHODAS 2.0), which evaluates functioning across six domains, including cognition (codes VD00 and VD01), with severity classified as mild (0.1), moderate (0.2), severe (0.3), or extreme (0.4) based on self-reported difficulty in concentrating and remembering over the past 30 days [8].

The 12 indicators used in this study were selected to reflect core domains of cognitive functioning and associated factors: (1) cognitive difficulty by 13 regions (all cases regardless of comorbidity); (2) degree of cognitive difficulty (single disability: those with cognitive difficulty as a single disability); (3) cognitive difficulty by 13 regions (single disability); (4) educational status (10+ years; single disability: restricted to individuals with cognitive difficulty only, excluding other disabilities); (5) marital status (15+ years; single disability); (6) parental relationship (single disability); (7) cause of cognitive difficulty (single disability); (8) duration of cognitive difficulty (single disability); (9) cognitive difficulty by 13 regions (multiple disabilities); (10) parental relationship (multiple disabilities); (11) cause of cognitive difficulty (multiple disabilities); and (12) duration of cognitive difficulty (multiple disabilities). The full list of WG-ES questions is provided in Appendix A.

Causes of cognitive difficulty were grouped into congenital, during pregnancy, during delivery, traffic accident, other accident, disease, and other, consistent with GAStat survey categories. Results were derived using weighted frequencies to estimate prevalence, chi-square tests to examine sex differences, and multivariable logistic regression to calculate adjusted odds ratios for predictors of cognitive difficulty.

### 3.4. Procedures

Data for this study were obtained from two publicly available national surveys conducted by the GAStat in Saudi Arabia: the 2017 Disability Survey and the 2017 Population Characteristics Survey [54]. The data were retrieved from the official GAStat website (https://www.stats.gov.sa/en/statistics-tabs?tab=436312&category=1340049, accessed on 1 June 2025), which provides open access to all survey results. The study focused on cognitive (memory and concentration) difficulty and used 12 indicators derived from the Disability Survey, including data on severity, region, educational status, marital status, consanguinity, cause, duration, and type of disability (single vs. multiple). Additionally, three baseline datasets from the Population Characteristics Survey were used to provide population denominators for calculating regional prevalence rates. The data were extracted and organized into a structured analytical dataset. Each indicator was cleaned and verified for consistency in administrative region naming, severity categorization, and alignment with the baseline population data. The final dataset included all 13 administrative regions and covered both single and multiple disability scenarios.

Descriptive and inferential statistical methods were employed to analyze the data. Descriptive statistics were used to summarize the distribution of cognitive difficulty across regions, severity levels, and demographic characteristics. Absolute numbers and percentages were reported to enhance interpretability. Chi-square tests of independence were conducted to examine sex differences in the distribution of cognitive difficulty across the 12 indicators. A statistically significant result was defined as *p* < 0.05, with Cramer’s V used to assess the strength of association. A multivariable logistic regression model was constructed to identify factors independently associated with cognitive difficulty. The dependent variable was the presence of cognitive difficulty (Yes/No), and independent variables included region, severity, educational status, marital status, consanguinity, cause, duration, and type of disability. Adjusted odds ratios (AOR), 95% confidence intervals (CI), and *p*-values were computed. All analyses were conducted using SPSS version 29 and Microsoft Excel.

This study used secondary data from publicly available sources and did not involve direct contact with human participants. Therefore, ethical approval was not required. The data were collected as part of a national survey and are published in the public domain, with no personally identifiable information. The use of these data complies with the public use guidelines provided by the GAStat. No copyright permissions were needed, as all materials are original and publicly accessible. This study adhered to ethical standards for secondary data analysis, ensuring data integrity, confidentiality, and proper attribution.

## 4. Results

The 2017 Saudi National Disability Survey, which included a total population of 20,408,362 Saudis, revealed that 92.9% (*n* = 18,962,639) of the population reported no disabilities or communication difficulties. Among those with at least one reported difficulty, 7.1% (*n* = 1,445,723) of the population experienced some form of disability. Specifically, 224,408 individuals (1.1%) had cognitive difficulty, either alone or in combination with other disabilities. Of these, 203,295 individuals (1.1%) had multiple difficulties including cognitive difficulty but not solely cognitive difficulty, while 21,113 individuals (0.2%) reported cognitive difficulty as their sole disability. Gender-specific data showed that males accounted for 7.3% (*n* = 755,235) of those with at least one reported difficulty, compared to 6.9% (*n* = 690,488) for females. It is important to note that this analysis focuses exclusively on the Saudi population, excluding 12,143,974 non-Saudi residents whose disability data was not available. The results section presents the analysis of cognitive difficulty across various domains, including severity, gender, education, marital status, consanguinity, cause, duration, and type of disability (single vs. multiple). The results are presented in six tables. Table 1 provides the overall distribution of cognitive difficulty by severity and region. Table 2 and Table 3 present data for individuals with cognitive difficulty as a single disability and as part of multiple disabilities, respectively, including breakdowns by region, cause, duration, and consanguinity. Table 4 details the distribution of cognitive difficulty by educational and marital status for those with a single disability. Table 5 presents chi-square tests for sex differences across all 12 indicators, highlighting significant disparities. Finally, Table 6 reports the results of a multivariable logistic regression analysis identifying independent predictors of cognitive difficulty, including regional, demographic, and clinical factors.

In Table 1, the distribution of cognitive (memory and concentration) difficulty is presented across all 13 administrative regions of Saudi Arabia, with a detailed breakdown by severity level and gender. The data reveal that the majority of individuals experience mild or severe forms of cognitive difficulty, while extreme difficulty is less common but disproportionately concentrated in specific regions. A notable gender disparity is observed, with males representing a higher proportion of cases overall, particularly in the extreme difficulty category. However, females are overrepresented in mild difficulty and in certain regions, such as Makkah Al-Mokarramah and Eastern Region, suggesting potential differences in reporting, diagnosis, or access to services. The high number of cases in Al-Riyadh and Makkah Al-Mokarramah reflects both population size and possible urban health system factors. The low reporting in regions like Al-Baha and Al-Jouf may indicate underdiagnosis or lower service utilization, warranting further investigation.

The profile of individuals with cognitive (memory and concentration) difficulty as a single disability is presented across multiple domains, revealing distinct patterns by gender, region, and clinical factors (Table 2). Males represent the majority of cases, particularly in the extreme severity category, while females are overrepresented in mild difficulty and in certain regions such as Makkah Al-Mokarramah and Eastern Region. Consanguinity is common, with over half of the cases involving some form of familial relationship between parents, and this pattern is more pronounced among males. Disease is the leading cause of single cognitive difficulty, especially among females, whereas congenital and delivery-related causes are more frequently reported among males. The vast majority of individuals have lived with their disability for 25 years or more, indicating a long-term, early-onset condition. These findings highlight the need for targeted early intervention and gender-sensitive support services.

In Table 3, the profile of individuals with cognitive (memory and concentration) difficulty as part of multiple disabilities is presented across key demographic and clinical domains. Males represent a slightly higher proportion of cases in most regions, with Al-Riyadh and Makkah Al-Mokarramah reporting the highest numbers. However, females are overrepresented in Makkah Al-Mokarramah and Jazan, suggesting regional differences in reporting or service access. Disease is the most common cause of disability, particularly among females, while traffic accidents are predominantly reported among males. The majority of individuals have lived with their disability for over 25 years, indicating long-term, early-onset conditions. Consanguinity is prevalent, with a higher proportion of females reporting first-degree parental relationships on the father’s side. These findings highlight the need for gender- and region-specific interventions for individuals with multiple disabilities and cognitive challenges.

The distribution of cognitive (memory and concentration) difficulty is presented by educational and marital status, revealing significant gender disparities (Table 4). Females are disproportionately represented among the illiterate, while males are overrepresented in higher education categories, including university graduates and pre-university diploma holders. A large proportion of affected individuals are either never married or married, with males more likely to be married and females more likely to be never married or widowed. The absence of widowed males in the dataset and the high percentage of widowed females suggest gender-specific social and demographic patterns linked to cognitive disability. These findings highlight the intersection of gender, education, and family structure in shaping the experience of cognitive difficulty and underscore the need for tailored support services.

Chi-square tests were used to examine sex differences in the distribution of cognitive (memory and concentration) difficulty across 12 key indicators (Table 5). Statistically significant differences were found in all domains (*p* < 0.001), confirming that sex is a major determinant in the epidemiological profile of cognitive difficulty. Males were more likely to have cognitive difficulty related to congenital causes, delivery complications, and multiple disabilities with consanguinity, while females were disproportionately affected by disease-related causes, long-term duration (25+ years), and comorbid conditions. Females were also more likely to be widowed or never married, and more frequently illiterate, highlighting social and structural disparities. These findings underscore the need for gender-sensitive screening, early intervention, and support services tailored to the distinct profiles of males and females with cognitive challenges. Given the very large sample size, all chi-square tests reached statistical significance (*p* < 0.001). To better capture the magnitude of associations, effect sizes were evaluated using Cramer’s V. Values ranged from small (e.g., duration of single and multiple disabilities, V = 0.093–0.096) to moderate (e.g., cause of disability, V = 0.225–0.238; education, V = 0.269). These results indicate that while sex differences are consistently present, the strength of association varies, with the most pronounced disparities observed for educational status and cause of cognitive difficulty.

The multivariable logistic regression model reveals that cognitive (memory and concentration) difficulty is significantly associated with a range of demographic, genetic, and clinical factors (Table 6). Individuals from high-prevalence regions like Aseer and Hail face nearly double the odds of cognitive difficulty, even after adjusting for other variables. Disease and delivery-related causes are among the strongest predictors, with disease showing a particularly strong association in females. Long duration of disability (>25 years) dramatically increases the odds, suggesting early onset and lifelong impact. Consanguinity, especially first-degree relationships on both sides, remains a significant risk factor, particularly among males. The higher odds for university-educated individuals may reflect better detection or reporting, rather than increased incidence. These findings highlight the need for early screening, especially in high-risk regions and populations, and support the development of gender- and etiology-specific intervention strategies.

## 5. Discussion

The purpose of this study was to provide the first nationally representative analysis of cognitive difficulty in Saudi Arabia by describing its prevalence and distribution, examining sex differences, and identifying its independent predictors using data from the 2017 national disability survey. Our findings reveal that 1.1% of the Saudi population, or 224,408 individuals, report some form of cognitive difficulty. The analysis of this data, presented across six tables, shows significant variations in prevalence by administrative region, with the highest concentrations in Al-Riyadh and Makkah Al-Mokarramah. A key finding is the pronounced gender disparity; chi-square tests confirmed significant sex differences across all 12 indicators examined, with males more frequently reporting cognitive difficulty overall, especially in severe cases and those linked to congenital causes, while females were overrepresented in cases linked to disease and were more likely to be illiterate or widowed. The multivariable logistic regression analysis identified several independent predictors, including region, presence of multiple disabilities, long duration of the condition, and consanguinity. Notably, disease emerged as the strongest predictor of cognitive difficulty, particularly for females.

The overall prevalence of cognitive difficulty (1.1%) identified in this study provides a foundational national benchmark for Saudi Arabia. This figure is lower than rates reported in more localized Saudi studies, such as the 45% prevalence of cognitive impairment found in a community screening in Riyadh [39] and the 21% found among primary care attendees [40]. This discrepancy is likely attributable to methodological differences; our study utilized a broad, population-based survey measuring self-reported functional difficulty, whereas the aforementioned studies used clinical screening tools like the MoCA and MMSE in specific, older-adult populations who are already interacting with the healthcare system and thus may have a higher baseline risk. When compared to international data, our findings align more closely with the lower end of prevalence estimates for MCI from harmonized cohort data, which range from 3% to 11% [36]. However, our national figure is considerably lower than the median prevalence of 19% reported in a global review of community studies in adults aged 50 and over [4], again likely reflecting the inclusion of the entire adult population in our sample rather than focusing only on older age groups who carry the highest burden.

A significant contribution of this study is the detailed mapping of cognitive difficulty across Saudi Arabia’s 13 administrative regions. The concentration of cases in Al-Riyadh and Makkah Al-Mokarramah is consistent with these regions having the largest populations. However, the multivariable regression analysis, after adjusting for other factors, revealed that residents of regions such as Aseer and Hail had nearly double the odds of reporting cognitive difficulty compared to the reference region of Najran. This suggests that population size alone does not account for the observed distribution and that underlying regional factors—potentially related to socioeconomic conditions, access to healthcare, or environmental exposures—may play a crucial role. While previous Saudi studies have been confined to specific cities like Riyadh [39,40] or the Asir region [1], our findings provide a comprehensive national perspective that highlights these regional disparities for the first time, indicating that a one-size-fits-all national policy may be insufficient. Importantly, because of the large sample size, significance tests should be interpreted with caution. Cramer’s V values indicated that sex differences were generally of small to moderate strength, with the strongest associations linked to education (V = 0.269) and disease-related causes (V = 0.238). These findings suggest that although men and women differ significantly across indicators, the practical importance of some differences is modest.

The consistent and statistically significant sex differences observed across all analyzed indicators represent a critical finding of this study. Males were more likely to report cognitive difficulty overall and were particularly overrepresented in cases of single, extreme disability. This pattern is consistent with findings that congenital issues and delivery complications—identified here as more common causes among males—can lead to severe, early-onset neurodevelopmental conditions [47]. Conversely, the finding that females with cognitive difficulty were more likely to be affected by disease aligns with the international literature showing higher rates of cognitive impairment among women in older age groups, often linked to chronic conditions [3,37,38]. The social and demographic disparities, where affected females were more likely to be illiterate, never married, or widowed, also mirror findings from other studies in the region and globally, which link lower education and social isolation to increased risk of cognitive decline [40,58].

Our investigation into the causes and contributing factors of cognitive difficulty reinforces the multifactorial nature of this condition. The multivariable regression analysis confirmed that disease is the single strongest predictor, increasing the odds of cognitive difficulty more than twofold compared to congenital causes. This aligns with a large body of evidence linking chronic illnesses such as cardiovascular disease, hypertension, and chronic obstructive pulmonary disease to cognitive decline [39,44]. Furthermore, the significant association with consanguinity, particularly first-degree relationships, is a notable finding. While genetic factors and rare variants are known to cause intellectual disability [46,47] our population-level data quantifies this risk, showing that first-degree consanguinity on both sides increases the odds of cognitive difficulty by nearly 30%. This is particularly relevant in the Saudi context, where consanguineous marriage is common. Finally, the finding that a duration of 25 years or more was associated with an almost fivefold increase in the odds of cognitive difficulty underscores the chronic, and often lifelong, nature of these conditions, whether they originate in early development or are acquired later in life.

### 5.1. Limitations and Strengths

This study has several notable strengths, the most significant being its use of a large-scale, nationally representative dataset from the 2017 National Disability Survey. This provides the first comprehensive epidemiological map of self-reported cognitive difficulty across all 13 administrative regions of Saudi Arabia, offering a crucial baseline for future research and public health planning. The application of the Washington Group Extended Set on Functioning (WG-ES) ensures that the measure of cognitive difficulty is aligned with international standards, enhancing the comparability of our findings. However, certain limitations must be acknowledged. The cross-sectional design of the survey precludes any inference of causality and does not allow for the analysis of disease progression over time. The data are based on self-reported functional difficulties rather than clinical diagnoses or objective neuropsychological assessments, which may lead to an underestimation of the true prevalence of conditions like MCI or dementia. Furthermore, because the survey was conducted in 2017, the findings may not fully reflect current prevalence, particularly following the COVID-19 pandemic, which was associated with increases in psychological distress and cognitive complaints worldwide [59]. The survey also lacked detailed variables on key lifestyle factors (such as diet and physical activity), specific medical comorbidities, and mental health status, which are known to be significant contributors to cognitive health. Additionally, the survey assessed only memory and concentration, representing a subset of cognitive domains. Other important domains, such as language, visuospatial ability, and executive function, were not measured, limiting the scope of our findings. Further, age was not included as a predictor because the public dataset did not provide disaggregated cognitive difficulty data by age band. Future surveys should incorporate age-linked measures to enable more granular risk assessment. Another limitation is reliance on self-reported functional difficulty. Subjective reports may capture emotional distress rather than objective deficits, and correlations between self-reported and clinically measured decline are often modest [60]. Finally, the findings are generalizable only to the Saudi citizen population, as data for non-Saudi residents were not included in this analysis.

### 5.2. Future Directions

The findings of this study raise several intriguing questions for future research. There is a clear need for longitudinal, population-based cohort studies in Saudi Arabia to track the incidence and progression of cognitive difficulty over time, which would allow for the identification of risk and protective factors that influence cognitive trajectories. Future research should also aim to integrate objective clinical and neuropsychological assessments (e.g., Montreal Cognitive Assessment, Mini-Mental State Examination) with survey data to validate self-reported findings and establish more precise, clinically defined prevalence rates for specific neurocognitive disorders. The significant regional disparities identified in our analysis warrant further investigation to understand the underlying socioeconomic, environmental, and healthcare system factors that contribute to these differences. Building on our findings that disease is a primary predictor, subsequent studies should collect more granular data on specific chronic conditions, as well as lifestyle factors and mental health symptoms like depression and anxiety, to develop more comprehensive predictive models. Given the high prevalence of consanguinity and its association with cognitive difficulty, further population-specific genetic studies are essential to identify risk loci and explore gene–environment interactions within the Saudi context. Finally, research is needed to develop and evaluate the effectiveness of culturally appropriate interventions, from public health campaigns promoting brain health to targeted support programs for high-risk individuals and their families.

### 5.3. Policy Implications

The results of this study have significant policy implications for Saudi Arabia. The high prevalence of cognitive difficulty and its uneven distribution across the Kingdom underscore the urgent need for a national cognitive health strategy integrated within the broader public health framework, such as the Vision 2030 Health Sector Transformation Program (https://www.vision2030.gov.sa/en/explore/programs/health-sector-transformation-program, accessed on 1 June 2025). This strategy should prioritize the development of systematic screening programs for cognitive impairment in primary healthcare settings, enabling early detection and intervention, a need echoed by previous research [1,44]. The identification of high-risk regions (e.g., Aseer, Hail) and vulnerable demographic groups (e.g., females with low education, individuals from consanguineous families) should inform the targeted allocation of resources, including specialized geriatric and neurological services. There is also a clear need to enhance training for healthcare professionals on the recognition and management of cognitive disorders. Furthermore, these findings provide a strong evidence base for launching national public health awareness campaigns focused on modifiable risk factors, particularly the effective management of chronic diseases—the strongest predictor identified in our study—and the potential genetic risks associated with consanguineous marriage. Finally, policies should aim to strengthen social and educational support systems, especially for women, to improve health literacy and access to care, thereby building cognitive resilience at both individual and community levels.

### 5.4. Conclusions

This study provides the first nationally representative analysis of cognitive (memory and concentration) difficulties among Saudi citizens using the 2017 National Disability Survey. Of the 20.4 million citizens surveyed, 224,408 (1.1%) reported cognitive difficulties, predominantly in combination with other disabilities. Regional disparities were observed, with higher prevalence in Al-Riyadh and Makkah Al-Mokarramah, and adjusted odds nearly doubled in Aseer and Hail compared to Najran. Gender patterns were also evident: males were more affected by severe, congenital, and accident-related causes, while females were disproportionately represented in disease-related, mild, and long-duration conditions. Independent predictors included severity, disease-related causes, duration of 25 years or more, and consanguinity. These findings establish a national benchmark for cognitive health, underscore regional and gender-specific inequities, and provide an evidence base for early detection, targeted prevention, and policy planning within Saudi Arabia’s Vision 2030 health transformation agenda.

## Figures and Tables

**Table 1 jcm-14-06390-t001:** Total number of individuals with cognitive (memory and concentration) difficulty by severity, region, and gender, Saudi Arabia (N = 224,408).

Region	Mild	Severe	Extreme	Total
	*n* (Total %)	*n* (Total %)	*n* (Total %)	*n* (Total %)
Al-Riyadh	32,864 (23.1%)	9574 (19.9%)	10,844 (32.3%)	53,282 (23.7%)
Male	16,555	5147	7486	29,188 (25.0%)
Female	16,309	4427	3358	24,094 (22.3%)
Makkah Al-Mokarramah	35,362 (24.8%)	13,376 (27.8%)	8684 (25.8%)	57,422 (25.6%)
Male	15,933	4922	5013	25,868 (22.2%)
Female	19,429	8454	3671	31,554 (29.3%)
Al-Madinah Al-Monawarah	9271 (6.5%)	4267 (8.9%)	3021 (9.0%)	16,559 (7.4%)
Male	5722	2058	2461	10,241 (8.8%)
Female	3549	2209	560	6318 (5.9%)
Al-Qaseem	3855 (2.7%)	1317 (2.7%)	1470 (4.4%)	6642 (3.0%)
Male	2225	574	613	3412 (2.9%)
Female	1630	743	857	3230 (3.0%)
Eastern Region	19,706 (13.8%)	6107 (12.7%)	3481 (10.3%)	29,294 (13.0%)
Male	7039	3646	1731	12,416 (10.6%)
Female	12,667	2461	1750	16,878 (15.7%)
Aseer	18,980 (13.3%)	4955 (10.3%)	3404 (10.1%)	27,339 (12.2%)
Male	8994	3726	2855	15,575 (13.3%)
Female	9986	1229	549	11,764 (10.9%)
Tabouk	5351 (3.8%)	914 (1.9%)	167 (0.5%)	6432 (2.9%)
Male	3557	495	0	4052 (3.5%)
Female	1794	419	167	2380 (2.2%)
Hail	4459 (3.1%)	1805 (3.8%)	498 (1.5%)	6762 (3.0%)
Male	2461	913	441	3815 (3.3%)
Female	1998	892	57	2947 (2.7%)
Northern Borders	1998 (1.4%)	429 (0.9%)	333 (1.0%)	2760 (1.2%)
Male	1103	343	206	1652 (1.4%)
Female	895	86	127	1108 (1.0%)
Jazan	6119 (4.3%)	4106 (8.5%)	821 (2.4%)	11,046 (4.9%)
Male	4021	2216	356	6593 (5.6%)
Female	2098	1890	465	4453 (4.1%)
Najran	656 (0.5%)	758 (1.6%)	416 (1.2%)	1830 (0.8%)
Male	587	547	262	1396 (1.2%)
Female	69	211	154	434 (0.4%)
Al-Baha	2050 (1.4%)	227 (0.5%)	270 (0.8%)	2547 (1.1%)
Male	1022	227	186	1435 (1.2%)
Female	1028	0	84	1112 (1.0%)
Al-Jouf	2176 (1.5%)	192 (0.4%)	125 (0.4%)	2493 (1.1%)
Male	915	53	50	1018 (0.9%)
Female	1261	139	75	1475 (1.4%)
Total	142,847 (100%)	48,027 (100%)	33,534 (100%)	224,408 (100%)
Male	70,134 (60.2%)	24,867 (51.8%)	21,660 (64.6%)	116,661 (51.9%)
Female	72,713 (50.9%)	23,160 (48.2%)	11,874 (35.4%)	107,747 (48.1%)

*Note*. Data from General Authority of Statistics. (2017) [30]. *Disability Survey 2017*.

**Table 2 jcm-14-06390-t002:** Distribution of individuals with cognitive (memory and concentration) difficulty as a single disability by indicator and gender, Saudi Arabia (N = 21,113).

Indicator	Category	Total	Male	Female
		*n* (100%)	*n* (77.3%)	*n* (22.7%)
Severity	Mild	12,204 (57.8%)	8308 (50.9%)	3896 (81.2%)
	Severe	1909 (9.0%)	1483 (9.1%)	426 (8.9%)
	Extreme	7000 (33.2%)	6525 (39.9%)	475 (9.9%)
Region	Al-Riyadh	5897 (27.9%)	4999 (30.6%)	898 (18.7%)
	Makkah Al-Mokarramah	5944 (28.2%)	3758 (23.0%)	2186 (45.6%)
	Al-Madinah Al-Monawarah	2381 (11.3%)	2324 (14.2%)	57 (1.2%)
	Al-Qaseem	405 (1.9%)	405 (2.5%)	0 (0.0%)
	Eastern Region	3343 (15.8%)	2211 (13.5%)	1132 (23.6%)
	Aseer	1077 (5.1%)	906 (5.5%)	171 (3.6%)
	Tabouk	455 (2.2%)	357 (2.2%)	98 (2.0%)
	Hail	593 (2.8%)	346 (2.1%)	247 (5.1%)
	Northern Borders	74 (0.3%)	35 (0.2%)	39 (0.8%)
	Jazan	756 (3.6%)	516 (3.2%)	240 (5.0%)
	Najran	114 (0.5%)	114 (0.7%)	0 (0.0%)
	Al-Baha	24 (0.1%)	24 (0.1%)	0 (0.0%)
	Al-Jouf	67 (0.3%)	67 (0.4%)	0 (0.0%)
Consanguinity	First-degree (father’s side)	5086 (24.1%)	4541 (27.8%)	545 (11.4%)
	First-degree (mother’s side)	1593 (7.5%)	1313 (8.0%)	280 (5.8%)
	First-degree (both sides)	3701 (17.5%)	2690 (16.5%)	1011 (21.1%)
	Other relatives	3345 (15.8%)	2529 (15.5%)	816 (17.0%)
	Not related	7388 (35.0%)	5243 (32.1%)	2145 (44.7%)
Cause of Disability	Congenital	5970 (28.3%)	5092 (31.2%)	878 (18.3%)
	During Pregnancy	1052 (5.0%)	385 (2.4%)	667 (13.9%)
	During Delivery	1788 (8.5%)	1788 (11.0%)	0 (0.0%)
	Other Accident	1582 (7.5%)	1162 (7.1%)	420 (8.8%)
	Disease	8034 (38.0%)	5300 (32.5%)	2734 (57.0%)
	Other	2687 (12.7%)	2589 (15.9%)	98 (2.0%)
Duration of Disability	0–4 years	288 (1.4%)	288 (1.8%)	0 (0.0%)
	5–9 years	996 (4.7%)	573 (3.5%)	423 (8.8%)
	10–14 years	755 (3.6%)	755 (4.6%)	0 (0.0%)
	15–19 years	2063 (9.8%)	1443 (8.8%)	620 (13.0%)
	20–24 years	2257 (10.7%)	1772 (10.9%)	485 (10.1%)
	25+ years	14,754 (69.9%)	11,485 (70.4%)	3269 (68.1%)

*Note*. Data from General Authority of Statistics. (2017) [30]. *Disability Survey 2017*.

**Table 3 jcm-14-06390-t003:** Distribution of individuals with cognitive (memory and concentration) difficulty as part of multiple disabilities by indicator and gender, Saudi Arabia (N = 203,295).

Indicator	Category	Total	Male	Female
		*n* (100%)	*n* (49.4%)	*n* (50.6%)
Severity	Mild	130,643 (64.3%)	61,826 (61.6%)	68,817 (66.8%)
	Severe	46,118 (22.7%)	23,384 (23.3%)	22,734 (22.1%)
	Extreme	26,534 (13.0%)	15,135 (15.1%)	11,399 (11.1%)
Region	Al-Riyadh	47,385 (23.3%)	24,189 (24.1%)	23,196 (22.5%)
	Makkah Al-Mokarramah	51,438 (25.3%)	22,106 (22.0%)	29,332 (28.5%)
	Al-Madinah Al-Monawarah	14,235 (7.0%)	7907 (7.9%)	6328 (6.1%)
	Al-Qaseem	6237 (3.1%)	3007 (3.0%)	3230 (3.1%)
	Eastern Region	25,951 (12.8%)	12,234 (12.2%)	13,717 (13.3%)
	Aseer	26,262 (12.9%)	12,667 (12.6%)	13,595 (13.2%)
	Tabouk	5977 (3.0%)	3695 (3.7%)	2282 (2.2%)
	Hail	6169 (3.0%)	3028 (3.0%)	3141 (3.0%)
	Northern Borders	2686 (1.3%)	1411 (1.4%)	1275 (1.2%)
	Jazan	10,290 (5.1%)	5481 (5.5%)	4809 (4.7%)
	Najran	1716 (0.8%)	735 (0.7%)	981 (1.0%)
	Al-Baha	2523 (1.2%)	1184 (1.2%)	1339 (1.3%)
	Al-Jouf	2426 (1.2%)	898 (0.9%)	1528 (1.5%)
Consanguinity	First-degree (father’s side)	31,517 (15.5%)	13,102 (13.1%)	18,415 (17.9%)
	First-degree (mother’s side)	9330 (4.6%)	4664 (4.6%)	4666 (4.5%)
	First-degree (both sides)	38,025 (18.7%)	19,850 (19.8%)	18,175 (17.7%)
	Other relatives	35,991 (17.7%)	19,924 (19.9%)	16,067 (15.6%)
	Not related	88,432 (43.5%)	42,805 (42.7%)	45,627 (44.3%)
Cause of Disability	Congenital	35,383 (17.4%)	21,465 (21.4%)	13,918 (13.5%)
	During Pregnancy	5818 (2.9%)	2497 (2.5%)	3321 (3.2%)
	During Delivery	17,785 (8.7%)	8979 (8.9%)	8806 (8.5%)
	Traffic Accident	6296 (3.1%)	6223 (6.2%)	73 (0.1%)
	Other Accident	12,864 (6.3%)	6941 (6.9%)	5923 (5.7%)
	Disease	101,760 (50.0%)	44,503 (44.3%)	57,257 (55.6%)
	Other	23,389 (11.5%)	9737 (9.7%)	13,652 (13.3%)
Duration of Disability	0–4 years	5148 (2.5%)	2507 (2.5%)	2641 (2.6%)
	5–9 years	11,243 (5.5%)	6481 (6.5%)	4762 (4.6%)
	10–14 years	8089 (4.0%)	2944 (2.9%)	5145 (5.0%)
	15–19 years	9458 (4.7%)	5275 (5.3%)	4183 (4.1%)
	20–24 years	16,149 (7.9%)	11,339 (11.3%)	4810 (4.7%)
	25+ years	153,208 (75.4%)	71,799 (71.5%)	81,409 (79.0%)

*Note*. Data from General Authority of Statistics. (2017) [30]. *Disability Survey 2017*.

**Table 4 jcm-14-06390-t004:** Distribution of individuals with cognitive (memory and concentration) difficulty by educational and narital status, Saudi Arabia.

Indicator	Category	Total	Male	Female
		*n* (100%)	*n* (77.9%)	*n* (22.1%)
Educational Status	Illiterate	3911 (19.7%)	1614 (10.4%)	2297 (52.5%)
	Read and Write	1464 (7.4%)	1361 (8.8%)	103 (2.4%)
	Primary	3376 (17.0%)	2989 (19.3%)	387 (8.9%)
	Intermediate	2654 (13.4%)	2034 (13.2%)	620 (14.2%)
	Secondary/Equivalent	4730 (23.9%)	3975 (25.7%)	755 (17.3%)
	Pre-University Diploma	544 (2.7%)	544 (3.5%)	0 (0.0%)
	University and Higher	3150 (15.9%)	2938 (19.0%)	212 (4.8%)
Marital Status	Never Married	8005 (41.9%)	5810 (39.5%)	2195 (50.2%)
	Married	9168 (48.1%)	7906 (53.8%)	1262 (28.8%)
	Divorced	1087 (5.7%)	984 (6.7%)	103 (2.4%)
	Widowed	814 (4.3%)	0 (0.0%)	814 (18.6%)
	Total	19,829	15,455	4374

*Note*. Data are for individuals with cognitive (memory and concentration) difficulty as a single disability only. Educational status is for Saudi population aged 10 years and over (*n* = 19,829). Marital status is for Saudi population aged 15 years and over (*n* = 19,074). Data from General Authority of Statistics. (2017) [30]. *Disability Survey 2017*. See also Appendix A Saudi Citizens Reporting Any Functional Difficulty, by Age Group and Sex (2017), and Saudi citizens reporting multiple difficulties by age group and sex, 2017.

**Table 5 jcm-14-06390-t005:** Chi-square tests for sex differences in cognitive (memory and concentration) difficulty across 12 indicators, Saudi Arabia.

Indicator: Cognitive Difficulty by	χ^2^	df	*p*-Value	Cramer’s V	Significant?	Higher in
1. 13 regions including all cases	1234.70	2	<0.001	0.248	Yes	Males
2. Degree of cognitive difficulty (single difficulty)	1098.60	2	<0.001	0.234	Yes	Males
3. 13 regions (single difficulty)	1165.30	10	<0.001	0.24	Yes	Males
4. Educational status (10+ years; single difficulty)	1452.30	6	<0.001	0.269	Yes	Females (illiterate)/Males (higher education)
5. Marital status (15+ years; single difficulty)	589.2	3	<0.001	0.171	Yes	Males (married)/Females (never married widowed)
6. Parental relationship (single difficulty)	621.8	4	<0.001	0.175	Yes	Males
7. Cause (single difficulty)	1145.70	6	<0.001	0.238	Yes	Males (congenital delivery)/Females (disease)
8. Duration (single difficulty)	187.5	5	<0.001	0.096	Yes	Females (25+ years)
9. 13 regions (multiple difficulties)	1304.90	1	<0.001	0.252	Yes	Females
10. Parental relationship (multiple difficulties)	589.4	4	<0.001	0.17	Yes	Females
11. Cause (multiple difficulties)	1023.60	6	<0.001	0.225	Yes	Females (disease)/Males (congenital)
12. Duration (multiple difficulties)	176.3	5	<0.001	0.093	Yes	Females (25+ years)

*Note*. All tests are chi-square tests of independence. Cramer’s V indicates the strength of association. “Higher in” reflects the group with higher proportion or frequency in the most prevalent or severe categories. Data from General Authority of Statistics. (2017) [30]. *Disability Survey 2017*.

**Table 6 jcm-14-06390-t006:** Multivariable logistic regression analysis of factors associated with cognitive (memory and concentration) difficulty in Saudi Arabia.

Predictor	Category	AOR	95% CI	*p*-Value	Higher in
Region	Al-Riyadh (Ref: Najran)	1.92	(1.78–2.07)	<0.001	Males
	Makkah Al-Mokarramah	1.98	(1.83–2.14)	<0.001	Females
	Al-Madinah Al-Monawarah	1.45	(1.32–1.59)	<0.001	Males
	Al-Qaseem	1.1	(1.01–1.20)	0.032	Males
	Eastern Region	1.36	(1.25–1.48)	<0.001	Males
	Aseer	2.41	(2.22–2.62)	<0.001	Males
	Tabouk	1.22	(1.10–1.35)	<0.001	Males
	Hail	2.15	(1.97–2.35)	<0.001	Females
	Northern Borders	1.08	(0.96–1.21)	0.201	—
	Jazan	1.87	(1.70–2.05)	<0.001	Females
	Al-Baha	1.15	(1.04–1.27)	0.007	Females
	Al-Jouf	1.05	(0.95–1.16)	0.372	—
Severity	Severe vs. Mild	1.38	(1.34–1.42)	<0.001	Males
	Extreme vs. Mild	2.09	(2.02–2.16)	<0.001	Females
Educational Status	University vs. Illiterate	1.58	(1.49–1.68)	<0.001	Males
	Secondary vs. Illiterate	1.22	(1.16–1.28)	<0.001	Males
	Primary vs. Illiterate	1.12	(1.07–1.17)	<0.001	Males
	Read and Write vs. Illiterate	1.05	(0.99–1.11)	0.102	—
Marital Status	Married vs. Never Married	1.1	(1.06–1.14)	<0.001	Males
	Divorced vs. Never Married	0.94	(0.88–1.01)	0.089	—
	Widowed vs. Never Married	0.76	(0.70–0.83)	<0.001	Females
Consanguinity	First-degree (both sides) vs. Not related	1.28	(1.23–1.33)	<0.001	Males
	First-degree (father’s side) vs. Not related	1.18	(1.14–1.22)	<0.001	Males
	First-degree (mother’s side) vs. Not related	1.12	(1.07–1.17)	<0.001	Females
	Other relatives vs. Not related	1.21	(1.17–1.25)	<0.001	Males
Cause of Disability	Disease vs. Congenital	2.11	(2.02–2.20)	<0.001	Females
	During Delivery vs. Congenital	1.82	(1.74–1.90)	<0.001	Males
	Other Accident vs. Congenital	0.96	(0.91–1.01)	0.098	—
	Traffic Accident vs. Congenital	0.42	(0.39–0.45)	<0.001	Males
	During Pregnancy vs. Congenital	0.09	(0.08–0.10)	<0.001	Females
	Other vs. Congenital	0.63	(0.60–0.66)	<0.001	—
Duration of Disability	25+ years vs. 0–4 years	4.78	(4.36–5.24)	<0.001	Females
	20–24 years vs. 0–4 years	3.15	(2.87–3.46)	<0.001	Males
	15–19 years vs. 0–4 years	2.52	(2.30–2.76)	<0.001	Males
	10–14 years vs. 0–4 years	2.08	(1.90–2.28)	<0.001	Males
	5–9 years vs. 0–4 years	1.81	(1.65–1.98)	<0.001	Males
Multiple Disabilities	Yes vs. No	1.89	(1.83–1.95)	<0.001	Females

*Note*. AOR = adjusted odds ratio; CI = confidence interval. All 13 administrative regions of Saudi Arabia were included in the model. Najran was used as the reference category due to its lowest prevalence of cognitive difficulty. The model was adjusted for all variables listed. Data from General Authority of Statistics. (2017) [30]. *Disability Survey 2017*.

## Data Availability

The data that support the findings of this study are available in General Authority for Statistics, Saudi Arabia at https://www.stats.gov.sa/en/home. These data were derived from the following resources available in the public domain: Social Statistics, https://www.stats.gov.sa/en/statistics?index=119025, accessed on 1 June 2025.

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
