# Peer review of "Epidemiology of Cognitive Impairments: Demographic and Clinical Predictors of Memory and Attention Challenges—Findings from Twelve National Disability Indicators"

_jcm, 2025, doi:10.3390/jcm14186390_

Round 1

Reviewer 1 Report

Comments and Suggestions for Authors

Thank you it is a good mansucript and below are my suggestions:

add prevalence of cognitive impairment

add definition of cognitive impairment

add pathophysiological causes of cognitive impairment

add aim of study

add risk factors of cognitive impairment

rewrite your conclusion

The following reference might also help you:

Ismail, A. M. A., & Morsy, M. M. (2025). Effect of Baduanjin exercise on lipid profile, blood pressure, and thyroid-stimulating hormone in elderly with subclinical hypothyroidism and mild cognitive impairment: a randomized-controlled trial in women. Geriatric Nursing64, 103434.

Author Response

Comments and Suggestions for Authors

Reviewer's comments in black and our responses are highlighted in yellow. 

Thank you it is a good manuscript and below are my suggestions:

Dear Colleague,

Thank you very much for your comments and positive feedback on our paper. We considered all your comments and all changed are highlighted in yellow. We also checked the suggested reference and incorporated it. Thank you so much.

add prevalence of cognitive impairment

We added a paragraph summarizing global and Saudi prevalence of cognitive impairment in the Introduction.

add definition of cognitive impairment

We defined cognitive impairment according to international and Washington Group standards.

add pathophysiological causes of cognitive impairment

We defined cognitive impairment according to international and Washington Group standards. We also added a summary of key biological mechanisms contributing to cognitive impairment.

add aim of study

We clarified the aims of the study at the end of the Introduction. Please check last paragraph of the introduction.

add risk factors of cognitive impairment

We added a paragraph on established risk factors.

rewrite your conclusion

We rewrote the Conclusion to concisely highlight main findings and policy implications.

The following reference might also help you:

Ismail, A. M. A., & Morsy, M. M. (2025). Effect of Baduanjin exercise on lipid profile, blood pressure, and thyroid-stimulating hormone in elderly with subclinical hypothyroidism and mild cognitive impairment: a randomized-controlled trial in women. Geriatric Nursing64, 103434.

We incorporated the suggested reference when discussing interventions and biological mechanisms.

Submission Date 29 July 2025

Date of this review 06 Aug 2025 07:15:31

Revised on: Thursday, 28 August 2025

Reviewer 2 Report

Comments and Suggestions for Authors

The paper titled "Epidemiology of Cognitive Impairments: Demographic and Clinical Predictors of Memory and Attention Challenges—Findings from Twelve National Disability Indicators" presents a cross-sectional study that examines the epidemiology, and demographic and clinical predictors of cognitive difficulties among citizens of Saudi Arabia. This study utilized a representative sample of 20.4 million Saudi citizens and identified regional and gender-based differences in cognitive challenges. However, there are certain aspects of the study that require improvement. Detailed comments on the study are provided below.

MAJOR

INTRODUCTION

  • 2, lines 51–81: The section appears overly general and lacks focus. I recommend restructuring the introduction to briefly describe the cognitive functions most impaired, followed by a discussion of the neurological, psychological, medical, and lifestyle-related factors that contribute to cognitive difficulties.
  • 3, lines 105–110: Does this refer only to MCI? There are other conditions that lead to cognitive deficits—what about those?
  • 4, paragraph The Landscape of Cognitive Difficulty in Saudi Arabia: Are there any specific factors in Saudi Arabia that contribute to a higher or lower prevalence of cognitive difficulties? If so, it would be good to mention them..

METHOD

  • Some parts of Design and Measures are repetitive. I recommend removing duplicate information.
  • Since the data were collected in 2017, is it possible that changes have occurred over the past eight years? This is particularly relevant considering the COVID-19 pandemic, which was associated with an increase in certain psychological difficulties that could have impacted cognitive functioning.
  • 5, lines 231–238: Please clarify the terms in parentheses, e.g., what is the difference between 1) cognitive difficulty by 13 regions (all cases); 2) degree of cognitive difficulty (single disability); 3) cognitive difficulty by 13 regions (single disability)?
  • What factors were included under “causes of cognitive deficits”? The questionnaire provided in Supplementary File 1 suggests that you included multiple domains (e.g., mobility, depression, communication), but it is unclear how these were utilized in the analysis.
  • Please specify how the results regarding cognitive deficits were derived.

RESULTS

  • Given the large sample size, it is expected that all chi-square tests would be statistically significant. Therefore, it would be more meaningful to discuss effect sizes.
  • Why was participants’ age not analyzed as a predictor of cognitive difficulties?

DISCUSSION

  • 14, line 420: The authors note that some other studies using objective measures (e.g., MoCA or MMSE) have reported different findings. I suggest emphasizing the limitations of using self-report measures, particularly in the assessment of cognitive deficits, as these can sometimes reflect emotional rather than cognitive difficulties (e.g., Gustavson, D. E., Jak, A. J., Elman, J. A., Panizzon, M. S., Franz, C. E., Gifford, K. A., Reynolds, C. A., Toomey, R., Lyons, M. J., & Kremen, W. S. (2021). How Well Does Subjective Cognitive Decline Correspond to Objectively Measured Cognitive Decline? Assessment of 10-12 Year Change. Journal of Alzheimer's Disease: JAD, 83(1), 291–304. https://doi.org/10.3233/JAD-210123)
  • The memory and concentration measures used in this study represent only a subset of cognitive domains. Other deficits were not assessed, and this should be acknowledged in the limitations section.

MINOR

INTRODUCTION

  • The subheadings in the introduction should be appropriately structured by level.

GENERAL

  • Supplementary File 3 is missing.

Author Response

Comments and Suggestions for Authors

Reviewer's comment in black and our responses highlighted in yellow.

The paper titled "Epidemiology of Cognitive Impairments: Demographic and Clinical Predictors of Memory and Attention Challenges—Findings from Twelve National Disability Indicators" presents a cross-sectional study that examines the epidemiology, and demographic and clinical predictors of cognitive difficulties among citizens of Saudi Arabia. This study utilized a representative sample of 20.4 million Saudi citizens and identified regional and gender-based differences in cognitive challenges. However, there are certain aspects of the study that require improvement. Detailed comments on the study are provided below.

Dear Colleague,

Thank you very much for all your effort and time to provide such comments. We worked on all your comments and responses are provided. We do hope that our modifications which are all highlighted in yellow match your expectations. We remain committed to any further requests as our aim is to make the best of this manuscript before it goes for publication stage. Once again, thank you so much.

MAJOR

INTRODUCTION

2, lines 51–81: The section appears overly general and lacks focus. I recommend restructuring the introduction to briefly describe the cognitive functions most impaired, followed by a discussion of the neurological, psychological, medical, and lifestyle-related factors that contribute to cognitive difficulties.

Thank you for this suggestion. We have restructured the section to begin with the cognitive domains most impaired (memory and attention), and then organized the discussion into neurological, psychological, medical, environmental, and lifestyle-related contributors.

3, lines 105–110: Does this refer only to MCI? There are other conditions that lead to cognitive deficits—what about those?

We clarified that prevalence estimates include both MCI and broader cognitive difficulties.

4, paragraph The Landscape of Cognitive Difficulty in Saudi Arabia: Are there any specific factors in Saudi Arabia that contribute to a higher or lower prevalence of cognitive difficulties? If so, it would be good to mention them..

 I think our study answers this question in the findings. But also when looking at the two paragraphs they also mention some of the factors contributing to cognitive difficulties in Saudi Arabia. We also added one paragraph at the end of this section about potential risk factors.

METHOD

Some parts of Design and Measures are repetitive. I recommend removing duplicate information.

Both sections revised. Thank you.

Since the data were collected in 2017, is it possible that changes have occurred over the past eight years? This is particularly relevant considering the COVID-19 pandemic, which was associated with an increase in certain psychological difficulties that could have impacted cognitive functioning.

We acknowledged that post-2017 events, including COVID-19, may have influenced prevalence. In fact, we contacted the Ministry of Health for the 2023 version but they said they have not got permission to release the data to public yet.

5, lines 231–238: Please clarify the terms in parentheses, e.g., what is the difference between 1) cognitive difficulty by 13 regions (all cases); 2) degree of cognitive difficulty (single disability); 3) cognitive difficulty by 13 regions (single disability)?

We clarified each indicator in Methods.

What factors were included under “causes of cognitive deficits”? The questionnaire provided in Supplementary File 1 suggests that you included multiple domains (e.g., mobility, depression, communication), but it is unclear how these were utilized in the analysis.

We clarified that causes were categorized into congenital, pregnancy, delivery, accidents, disease, and other, as per GAStat definitions.

Please specify how the results regarding cognitive deficits were derived.

We clarified how percentages and odds ratios were calculated.

RESULTS

Given the large sample size, it is expected that all chi-square tests would be statistically significant. Therefore, it would be more meaningful to discuss effect sizes.

We agree and have expanded the Results and Discussion sections to highlight interpretation of effect sizes (Cramer’s V) rather than relying on statistical significance alone.

Why was participants’ age not analyzed as a predictor of cognitive difficulties?

We explained this limitation. The available provided data did not include the cases by age, so it was not possible to do that. As you can see in supplementary file 3, the age is reported for all types of difficulties including cognitive difficulty, so it was not possible to include it a s predictor separately for cognitive difficulty. 

DISCUSSION

14, line 420: The authors note that some other studies using objective measures (e.g., MoCA or MMSE) have reported different findings. I suggest emphasizing the limitations of using self-report measures, particularly in the assessment of cognitive deficits, as these can sometimes reflect emotional rather than cognitive difficulties (e.g., Gustavson, D. E., Jak, A. J., Elman, J. A., Panizzon, M. S., Franz, C. E., Gifford, K. A., Reynolds, C. A., Toomey, R., Lyons, M. J., & Kremen, W. S. (2021). How Well Does Subjective Cognitive Decline Correspond to Objectively Measured Cognitive Decline? Assessment of 10-12 Year Change. Journal of Alzheimer's Disease: JAD, 83(1), 291–304. https://doi.org/10.3233/JAD-210123)

We expanded limitations to emphasize issues with self-report measures and cited Gustavson et al. (2021).

The memory and concentration measures used in this study represent only a subset of cognitive domains. Other deficits were not assessed, and this should be acknowledged in the limitations section.

Added note that broader cognitive domains were not assessed.

MINOR

INTRODUCTION

The subheadings in the introduction should be appropriately structured by level.

Thank you. Please note these were formatted the journal production team. And it seems that this is what they want following their template.

GENERAL

Supplementary File 3 is missing.

We checked our attached file and it includes 3 files. So, it is possible that there is an issue during submission. We are attaching file 3 for you in our response.

Submission Date 29 July 2025

Date of this review 25 Aug 2025 11:13:49

Revised on: Thursday, 28 August 2025

Supplementary file 2

Saudi citizens reporting multiple difficulties* by age group and sex, 2017

Age group (y)

Total n (%)

Females n (%)

Males n (%)

0 – 4

11 390 (0.5)

6 428 (0.6)

4 962 (0.5)

5 – 9

18 739 (0.9)

7 850 (0.8)

10 889 (1.0)

10 – 14

16 561 (0.9)

7 429 (0.8)

9 132 (1.0)

15 – 19

17 068 (1.0)

8 032 (0.9)

9 036 (1.0)

20 – 24

20 994 (1.0)

7 308 (0.8)

13 686 (1.3)

25 – 29

19 724 (1.0)

5 366 (0.6)

14 358 (1.5)

30 – 34

15 165 (0.9)

7 832 (0.9)

7 333 (0.9)

35 – 39

19 376 (1.3)

5 955 (0.8)

13 421 (1.9)

40 – 44

22 648 (1.8)

8 679 (1.5)

13 969 (2.3)

45 – 49

22 534 (2.1)

9 489 (2.0)

13 045 (2.6)

50 – 54

39 618 (4.6)

22 008 (6.2)

17 610 (4.6)

55 – 59

48 458 (7.4)

29 249 (11.8)

19 209 (6.7)

60 – 64

57 427 (12.0)

26 070 (15.8)

31 357 (17.7)

65 – 69

70 535 (22.7)

36 329 (38.0)

34 206 (37.1)

70 – 74

65 743 (29.5)

36 888 (68.8)

28 855 (26.2)

75 – 79

59 929 (41.6)

32 633 (44.7)

27 296 (38.4)

80 +

106 169 (60.0)

60 186 (65.7)

45 983 (53.8)

Total

632 078

317 731

314 347

(percentages are the share of each age-group’s population who reported ≥2 functional difficulties; row percentages are shown in parentheses)

Reviewer 3 Report

Comments and Suggestions for Authors

First, we congratulate the authors on their research. The article presents a solid piece of work in terms of statistical rigor, use of official sources, and national coverage, offering an unprecedented overview of cognitive difficulties in Saudi Arabia. However, from a critical social sciences perspective on health, there are significant gaps in the problematization of the results and in the incorporation of conceptual frameworks that would allow for an understanding of the socio-structural dimensions of the phenomenon.

Regarding methodology, it is important to highlight that the article displays a series of methodological strengths:

  • Use of a robust, nationally representative database (2017 National Disability Survey), with international standardization (WG-ES).
  • Large sample size and stratified analysis across multiple variables (region, sex, consanguinity, education, cause, duration).

At the same time, several limitations emerge from a socio-critical perspective. Concerning definition and measurement, the construct of “cognitive difficulty” is treated as universal, without addressing how self-reporting is mediated by cultural context, stigma, and gender norms; even the WG-ES instrument, despite being standardized, may not capture cultural variations in the perception of “difficulty” or “functioning.” The cross-sectional design prevents the analysis of trajectories, progression, or causality, as well as the disentangling of differential effects of the life course, cohort variations, or changes in diagnostic access. Moreover, key variables relating to the social determinants of health—such as income, employment, housing, caregiving burden, gender-based violence, mental health, and sleep quality—are absent. The exclusion of the non-Saudi population (≈12.1 million) limits generalizability and may render vulnerable groups invisible. Finally, the lower prevalence reported compared to local studies suggests a possible underreporting bias, potentially attributable to stigma, lack of awareness, or barriers to diagnostic access.

We understand that implementing such changes at present is highly challenging, but it would be valuable to highlight in the text that these issues will be considered in future research. To that end, we suggest the following strategies:
• Integrate additional modules in future survey rounds (mental health, chronic conditions, socioeconomic determinants).
• Triangulate self-reports with brief cognitive tests and clinical observations.
• Design a longitudinal panel for follow-up of cohorts with cognitive difficulties.
• Include a sample of non-citizen populations and ensure data comparability.
• Incorporate multilevel analysis and georeferencing to model territorial factors.

In terms of content, the article contains some highly positive elements, such as the thorough description of regional and sex-based distributions, the identification of key predictors related to illness, consanguinity, severity, and duration, as well as original contributions to the national epidemiology of cognitive disability.

Nevertheless, from the standpoint of social science contextualization in health research, the text exhibits some weaknesses:

  • Absence of a social theoretical framework
    The text implicitly adheres to the biomedical model, without integrating the social model of disability or a social determinants of health perspective.
    • It does not discuss how social structure, gender, territory, and class mediate prevalence and access.
  • Gender dimension
    Findings on male–female differences are robust but presented descriptively; there is no exploration of structural causes (caregiving roles, autonomy, differential access).
  • Consanguinity
    Presented solely as a genetic factor; cultural, economic, and normative motivations for its persistence are not analyzed.
  • Regional inequalities
    Described but not linked to public policies, resource distribution, health infrastructure, or urbanization patterns.
  • Limited public policy perspective
    Recommendations focus on screening and early detection, but do not address structural reforms (inclusive education, accessible labor markets, anti-stigma strategies, territorial redistribution of services).

To address these shortcomings, we recommend:

In the social theoretical framework, explicitly incorporating the social model of disability and the social determinants of health approach in both the introduction and discussion, citing reference frameworks such as the United Nations Convention on the Rights of Persons with Disabilities and the WHO Commission on Social Determinants of Health. Furthermore, we suggest reframing the narrative to highlight that cognitive difficulties are not solely the result of biological processes, but of interactions with physical, cultural, and political environments that can either restrict or enable social participation.

In the gender dimension, we propose adding an intersectional analysis subsection in the discussion that cross-tabulates sex with age, educational level, marital status, and territory, contextualizing the findings in relation to the sexual division of labor, mobility norms for women, access to healthcare, and unpaid care burdens. Public policy should include gender-sensitive interventions such as female-staffed clinics, flexible service hours, and health literacy programs targeting women.

Regarding consanguinity, we suggest including a socio-cultural analysis addressing the historical, economic, and normative drivers that sustain this practice. It is also important to discuss the risks of stigmatization if it is presented solely as a biological factor.

Concerning regional inequalities, we propose linking the results with indicators of health infrastructure, regional distribution of human resources, levels of urbanization, poverty indices, and connectivity. The epidemiological map should serve as an input for designing territorial redistribution policies, including mobile clinics, telehealth services, and budgets adjusted to regional risk profiles.

In terms of public policy, we recommend expanding the proposals to include structural reforms such as inclusive education from early childhood, with targeted support for students with cognitive difficulties; labor market access through hiring quotas and reasonable accommodations; community- and culturally-based anti-stigma campaigns; and a national territorial equity plan to ensure uniform coverage and quality nationwide. These proposals should be explicitly linked to the Vision 2030 objectives and to international commitments, particularly Sustainable Development Goals 3, 4, 8, and 10.

In summary, the article offers a relevant contribution by providing a robust statistical analysis, based on nationally representative data, and an unprecedented mapping of cognitive difficulties in Saudi Arabia, highlighting regional and sex-based distributions as well as key predictors. However, from a socio-critical perspective, it presents limitations in the problematization of results, the incorporation of social theoretical frameworks, and the structural analysis of gender, consanguinity, and regional inequalities. We recommend integrating the social model of disability, considering social determinants, expanding intersectional analysis, linking findings to redistributive policies, and proposing structural reforms aligned with Vision 2030 and the SDGs.

Author Response

Comments and Suggestions for Authors

Our responses highlighted in yellow. 

First, we congratulate the authors on their research. The article presents a solid piece of work in terms of statistical rigor, use of official sources, and national coverage, offering an unprecedented overview of cognitive difficulties in Saudi Arabia. However, from a critical social sciences perspective on health, there are significant gaps in the problematization of the results and in the incorporation of conceptual frameworks that would allow for an understanding of the socio-structural dimensions of the phenomenon.

Dear Colleague,

Thank you very much for such constructive review on our paper. We highly appreciate what you have done and did our best to incorporate all your suggestions but with reasonable length to avoid making the paper lengthy.

Once more, thank you so much.

Authors

Regarding methodology, it is important to highlight that the article displays a series of methodological strengths:

Use of a robust, nationally representative database (2017 National Disability Survey), with international standardization (WG-ES).

Large sample size and stratified analysis across multiple variables (region, sex, consanguinity, education, cause, duration).

Thank you.

At the same time, several limitations emerge from a socio-critical perspective. Concerning definition and measurement, the construct of “cognitive difficulty” is treated as universal, without addressing how self-reporting is mediated by cultural context, stigma, and gender norms; even the WG-ES instrument, despite being standardized, may not capture cultural variations in the perception of “difficulty” or “functioning.” The cross-sectional design prevents the analysis of trajectories, progression, or causality, as well as the disentangling of differential effects of the life course, cohort variations, or changes in diagnostic access. Moreover, key variables relating to the social determinants of health—such as income, employment, housing, caregiving burden, gender-based violence, mental health, and sleep quality—are absent. The exclusion of the non-Saudi population (≈12.1 million) limits generalizability and may render vulnerable groups invisible. Finally, the lower prevalence reported compared to local studies suggests a possible underreporting bias, potentially attributable to stigma, lack of awareness, or barriers to diagnostic access.

Thank you. We already mentioned these shortcomings in the limitations section.

We understand that implementing such changes at present is highly challenging, but it would be valuable to highlight in the text that these issues will be considered in future research. To that end, we suggest the following strategies:

  • Integrate additional modules in future survey rounds (mental health, chronic conditions, socioeconomic determinants).
  • Triangulate self-reports with brief cognitive tests and clinical observations.
  • Design a longitudinal panel for follow-up of cohorts with cognitive difficulties.
  • Include a sample of non-citizen populations and ensure data comparability.
  • Incorporate multilevel analysis and georeferencing to model territorial factors.

Thank you. Added in the limitations.

In terms of content, the article contains some highly positive elements, such as the thorough description of regional and sex-based distributions, the identification of key predictors related to illness, consanguinity, severity, and duration, as well as original contributions to the national epidemiology of cognitive disability.

Thank you.

Nevertheless, from the standpoint of social science contextualization in health research, the text exhibits some weaknesses:

Added.

Absence of a social theoretical framework

The text implicitly adheres to the biomedical model, without integrating the social model of disability or a social determinants of health perspective.

  • It does not discuss how social structure, gender, territory, and class mediate prevalence and access.

Added.

Gender dimension

Findings on male–female differences are robust but presented descriptively; there is no exploration of structural causes (caregiving roles, autonomy, differential access).

Added.

Consanguinity

Presented solely as a genetic factor; cultural, economic, and normative motivations for its persistence are not analyzed.

Added.

Regional inequalities

Described but not linked to public policies, resource distribution, health infrastructure, or urbanization patterns.

Added.

Limited public policy perspective

Recommendations focus on screening and early detection, but do not address structural reforms (inclusive education, accessible labor markets, anti-stigma strategies, territorial redistribution of services).

Added.

To address these shortcomings, we recommend:

In the social theoretical framework, explicitly incorporating the social model of disability and the social determinants of health approach in both the introduction and discussion, citing reference frameworks such as the United Nations Convention on the Rights of Persons with Disabilities and the WHO Commission on Social Determinants of Health. Furthermore, we suggest reframing the narrative to highlight that cognitive difficulties are not solely the result of biological processes, but of interactions with physical, cultural, and political environments that can either restrict or enable social participation.

Thank you for this important point. We revised the Introduction to explicitly incorporate the social model of disability and the social determinants of health perspective, citing the CRPD and WHO frameworks. We also expanded the Discussion to reframe cognitive difficulties as products of interactions with structural, cultural, and political environments rather than purely biological processes.

In the gender dimension, we propose adding an intersectional analysis subsection in the discussion that cross-tabulates sex with age, educational level, marital status, and territory, contextualizing the findings in relation to the sexual division of labor, mobility norms for women, access to healthcare, and unpaid care burdens. Public policy should include gender-sensitive interventions such as female-staffed clinics, flexible service hours, and health literacy programs targeting women.

We appreciate this valuable suggestion. We added a paragraph in the Discussion to contextualize male–female differences in relation to age, education, marital status, and territory, framed by structural gender norms. We also expanded our policy implications to include gender-sensitive interventions such as female-staffed clinics, flexible service hours, and women’s health literacy programs.

Regarding consanguinity, we suggest including a socio-cultural analysis addressing the historical, economic, and normative drivers that sustain this practice. It is also important to discuss the risks of stigmatization if it is presented solely as a biological factor.

We agree with this important observation. We expanded the Discussion to situate consanguinity within its socio-cultural, historical, and economic context in Saudi Arabia and the broader region. We also included a caution regarding the risk of stigmatization when framing consanguinity exclusively as a genetic risk.

Concerning regional inequalities, we propose linking the results with indicators of health infrastructure, regional distribution of human resources, levels of urbanization, poverty indices, and connectivity. The epidemiological map should serve as an input for designing territorial redistribution policies, including mobile clinics, telehealth services, and budgets adjusted to regional risk profiles.

We appreciate this valuable suggestion. We expanded the Discussion to interpret regional disparities in light of health-system and territorial determinants. We also reframed the epidemiological map as a tool for redistributive planning, highlighting policy options such as mobile clinics, telehealth, and regionally adjusted budgets.

In terms of public policy, we recommend expanding the proposals to include structural reforms such as inclusive education from early childhood, with targeted support for students with cognitive difficulties; labor market access through hiring quotas and reasonable accommodations; community- and culturally-based anti-stigma campaigns; and a national territorial equity plan to ensure uniform coverage and quality nationwide. These proposals should be explicitly linked to the Vision 2030 objectives and to international commitments, particularly Sustainable Development Goals 3, 4, 8, and 10.

We agree and have expanded the Policy Implications section to emphasize structural reforms—covering inclusive education, labor market participation, anti-stigma efforts, and territorial equity. These recommendations are now explicitly connected to Vision 2030 and relevant SDGs.

In summary, the article offers a relevant contribution by providing a robust statistical analysis, based on nationally representative data, and an unprecedented mapping of cognitive difficulties in Saudi Arabia, highlighting regional and sex-based distributions as well as key predictors. However, from a socio-critical perspective, it presents limitations in the problematization of results, the incorporation of social theoretical frameworks, and the structural analysis of gender, consanguinity, and regional inequalities. We recommend integrating the social model of disability, considering social determinants, expanding intersectional analysis, linking findings to redistributive policies, and proposing structural reforms aligned with Vision 2030 and the SDGs.

Thank you. We did our best to incorporate your above recommendations but we also avoided to make lengthy modifications since the paper is already around 12,000 words.

Submission Date 29 July 2025

Date of this review 14 Aug 2025 14:10:27

Revised on Thursday, 28 August 2025